# Preliminary Report on the Influence of Acute Inflammation on Adiponectin Levels in Older Inpatients with Different Nutritional Status

**DOI:** 10.3390/ijms25042016

**Published:** 2024-02-07

**Authors:** Jakub Husejko, Marcin Gackowski, Jakub Wojtasik, Dominika Strzała, Maciej Pesta, Katarzyna Mądra-Gackowska, Jarosław Nuszkiewicz, Alina Woźniak, Mariusz Kozakiewicz, Kornelia Kędziora-Kornatowska

**Affiliations:** 1Department of Geriatrics, Faculty of Health Science, L. Rydygier Collegium Medicum in Bydgoszcz, Nicolaus Copernicus University in Torun, Skłodowska-Curie 9 Street, 85-094 Bydgoszcz, Poland; kubahusejko@gmail.com (J.H.); doominika05@gmail.com (D.S.); maciejpesta2000@gmail.com (M.P.); katarzyna.madra@cm.umk.pl (K.M.-G.); markoz@cm.umk.pl (M.K.); kornelia.kornatowska@cm.umk.pl (K.K.-K.); 2Department of Toxicology and Bromatology, Faculty of Pharmacy, L. Rydygier Collegium Medicum in Bydgoszcz, Nicolaus Copernicus University in Torun, A. Jurasza 2 Street, 85-089 Bydgoszcz, Poland; marcin.gackowski@cm.umk.pl; 3Centre for Statistical Analysis, Nicolaus Copernicus University in Toruń, Chopina 12/18 Street, 87-100 Toruń, Poland; jwojtasik@umk.pl; 4Department of Medical Biology and Biochemistry, Faculty of Medicine, Ludwik Rydygier Collegium Medicum in Bydgoszcz, Nicolaus Copernicus University in Toruń, Karłowicza 24 Street, 85-092 Bydgoszcz, Poland; jnuszkiewicz@cm.umk.pl

**Keywords:** adiponectin, bacterial infection, inflammation, older inpatients, malnutrition, MNA, nutritional status, oxidative stress in infection

## Abstract

Inflammation can be triggered by a variety of factors, including pathogens, damaged cells, and toxic compounds. It is a biological response of the immune system, which can be successfully assessed in clinical practice using some molecular substances. Because adiponectin, a hormone released by adipose tissue, influences the development of inflammation, its evaluation as a potential measure of inflammation in clinical practice is justified. In the present contribution, statistical comparison of adiponectin concentration and selected molecular substances recognized in clinical practice as measures of inflammation were utilized to demonstrate whether adipose tissue hormones, as exemplified by adiponectin, have the potential to act as a measure of rapidly changing inflammation when monitoring older hospitalized patients in the course of bacterial infection. The study showed no statistically significant differences in adiponectin levels depending on the rapidly changing inflammatory response in its early stage. Interestingly, the concentration of adiponectin is statistically significantly higher in malnourished patients than in people with normal nutritional levels, assessed based on the MNA. According to the results obtained, adiponectin is not an effective measure of acute inflammation in clinical practice. However, it may serve as a biomarker of malnutrition in senile individuals.

## 1. Introduction

Inflammation is the natural way in which the immune system responds to harmful stimuli such as pathogens, damaged cells, toxic compounds, or irradiation. It acts by removing harmful substances and initiating the healing process. Inflammation is, therefore, a defense mechanism that is crucial to maintaining good health. Various factors, such as infection, tissue injury, or cardiac infarction, can trigger inflammation by causing tissue damage [1]. The inflammation process causes changes in the patient’s general clinical condition but is also reflected in the concentration levels of certain substances, which result, among other things, from processes related to oxidative stress. It is possible to successfully confirm the presence and monitor the course of the inflammatory reaction using several laboratory tests. In clinical practice, the levels of C-reactive protein (CRP), procalcitonin (PCT), and leukocytes (WBC) are usually determined for this purpose. 

CRP is an acute-phase protein, a pro-inflammatory substance, the production of which is intensified during inflammation in the body. It is synthesized in the liver in response to increased interleukin-6 (IL-6) levels, which contributes to the activation of CRP transcription genes. Increased CRP levels may occur in both acute and chronic inflammation. However, this condition is most frequently observed in the case of infectious diseases [2,3]. Procalcitonin is a protein produced as a result of the transformation of pre-procalcitonin with the participation of endopeptidases in thyroid C cells. It is a precursor of a hormone naturally occurring in the body—calcitonin. Under homeostatic conditions, PCT concentrations are very low (typically < 0.05 ng/mL). The level of PCT is reactively increased as a result of the presence of heightened amounts of substances circulating in the blood, i.e., endotoxins, cytokines—interleukin-1b (IL-1b), and interleukin-6 (IL-6) or tumor necrosis factor (TNF-). A sudden increase in this marker is usually caused by an acute inflammatory response, most often caused by a bacterial infection [4,5]. Leukocytes (WBC) are a population of blood cells that participate in the immune system’s response to the inflammatory reaction. A special role in acute inflammation is played by neutrophils, whose influx into the inflamed area is stimulated by pro-inflammatory cytokines, i.e., IL-6 or IL-1b. Normally, the level of white blood cells in the blood is <10 × 10^9^/L and a result exceeding this threshold is called leukocytosis. Leukocytosis with neutrophilia is the basis for suspecting an acute inflammatory reaction and suggests deepening of the diagnostics each time to detect its cause [6]. There are many examples of substances involved in the development of inflammation. Notably, uric acid is also more frequently measured in clinical practice. However, it does not serve as a marker for monitoring inflammatory changes. It has been shown that older people had atypically high CRP and Il-6 concentrations if the patient had hyperuricemia [7]. Therefore, when considering substances for their potential to monitor inflammation, it is worth taking into account their correlations with uric acid.

The process of inflammation is noticeably related to oxidative stress. During inflammation there are many substances produced, which also play a significant role in the oxidative stress. Higher concentrations of IL-6, IL-1b, or TNF-**α** can be noticed in the progress of both these cases. In general, inflammation and oxidative stress often occur simultaneously in the organism, which causes one of the processes to affect another, activating several biochemical pathways, using receptor mechanisms and effects of many chemicals on body cells. Reactive oxygen species (ROS), which are produced by the effect of oxidative stress, entail activation of the immune system. Immune cells, especially leukocytes, are activated and the course of inflammation in the organism begins. Consequence of the mentioned actions are pro-inflammation cytokine level upgrowth and an exacerbation of the inflammation [8].

Adipose tissue is a hormonally active organ that produces adipokines, which play essential roles, for instance, in the maintenance of whole-body energy and metabolic homeostasis at both organ and system levels, inflammation, obesity, and oxidative stress, which accompanies many age-related diseases. One of the most hormonally significant adipokines and the richest one in human plasma is adiponectin (APN). Numerous studies have demonstrated that it has insulin-sensitizing, anti-atherogenic, and anti-inflammatory effects. Furthermore, decreased serum levels of APN are associated with chronic inflammation of metabolic disorders, which include type 2 diabetes, obesity, and atherosclerosis. However, recent studies have shown that APN could have pro-inflammatory roles in patients with autoimmune diseases. Thus, it appears that adiponectin has both pro-inflammatory and anti-inflammatory effects, which indirectly suggests that adiponectin has different physiological roles based on the isoform and effector tissue [9]. Its production in excessive amounts by adipose tissue cells contributes to the development of unfavorable metabolic processes. APN stimulates the phosphorylation of the enzymes: adenosine monophosphate-activated protein kinase (AMPK) and acetyl-CoA carboxylase (ACC). This leads to the activation of key enzymes for oxidation of fatty acids. Additionally, it stimulates the functioning of the APN-AMPK-eNOS signaling pathway, as a result of which the production of free oxygen radicals (ROS) is intensified. Moreover, it has been shown that a low level of adiponectin is associated with an increased concentration of pro-inflammatory cytokines (including IL-6 and TNF-) in the blood, which favors the intensification of the inflammatory reaction and may be the basis of many diseases related to systemic inflammatory reaction, i.e., rheumatoid arthritis, type II diabetes, or systemic lupus erythematosus. Because adiponectin influences the development of chronic inflammation, its evaluation as a potential measure of inflammation in clinical practice is justified [9,10]. It should also be remembered that adiponectin increases under conditions of oxidative stress [11], which indicates its potential in measuring inflammation. 

Senile patients admitted to the hospital for acute medical illness have a high prevalence of multimorbidity, malnutrition, and cognitive impairments [12]. In the case of such a patient, it is often difficult to make a quick and accurate diagnosis, so it takes time to implement appropriate treatment, and very often the patient’s clinical condition worsens. Moreover, treating infections in older patients can be difficult as their symptoms and signs may not be specific, leading to both over and undertreatment. Additionally, they have a weaker immune response to infections, which can affect the kinetics of biomarkers of infection [13]. When interpreting admission biomarkers for older adults with infections, caution must be taken as their prognostic abilities are lower in comparison to younger adults [14,15]. For these reasons, searching for and validation of good biomarkers of disorders common in the senile population is imperative and may facilitate hospitalization, i.e., allow for faster diagnosis and application of effective treatment. Using infection markers such as PCT to guide personalized treatment decisions is particularly helpful for geriatric patients. PCT is a valuable biomarker for assessing the risk of septic complications and adverse outcomes in geriatric patients. It can also help make informed decisions about whether to use antibiotics or not. 

In human research, leptin and APN are the two most extensively studied biomarkers among different molecules [16]. Many studies confirmed negative correlations between adiponectin and inflammatory biomarkers. However, the inverse correlation between the two variables was almost non-existent in participants who were older than 55 years of age. Adiponectin may be a possible predictor, among other things, of endothelial dysfunction [17], renal dysfunction [18], multiple sclerosis course [19], metabolic syndrome [20], osteoporosis [21], or even hair loss severity in alopecia areata [22]. Interestingly, data from patients with viral infections (chronic hepatitis C virus infections) [23] and human experimental endotoxemia [24] suggest altered patterns of APN release in these conditions. There is increasing evidence suggesting that adiponectin has an anti-inflammatory effect on the lungs. However, further research is needed to clarify the mechanism and roles of the APN pathway in protecting against lung diseases, including fungal, bacterial, and viral infections [25]. 

The study aimed to demonstrate whether adiponectin, as a prominent representative of adipose tissue hormones, has the potential to act as a measure of rapidly changing inflammation in the course of bacterial infection.

## 2. Results

### 2.1. The Influence of Nutritional Status on Evaluated Parameters

Table 1 illustrates the descriptive statistics of putative biomarkers divided by the MNA. According to the results, the mean uric acid concentration in hospitalized patients with normal nutritional status based on the MNA (N = 24) was 5.467 mg/dL. In patients at increased risk of malnutrition (N = 32), the mean uric acid concentration was 6.206 mg/dL, while in malnourished patients (N = 7), the mean concentration was 8.043 mg/dL. Therefore, there is a noticeable tendency for the uric acid level to increase as the nutritional status deteriorates, but the study group is too small to treat this as a rule.

The average adiponectin concentration in patients with a nutritional level within the normal range was 50.230 µg/mL, in patients with an increased risk of malnutrition it was 67.788 µg/mL, and in malnourished patients had a value of 79.626 µg/mL. As in the case of uric acid, here too there is a possible tendency to increase the concentration of adiponectin as the nutritional level deteriorates that needs further confirmation.

Similarly, the concentrations of inflammatory parameters, i.e., WBC and CRP, did not differ significantly in groups with different nutritional status. The average WBC at the time of admission to the hospital in patients with normal nutritional status was 9.195 × 10^3^/µL, with the risk of malnutrition 10,295 × 10^3^/µL, and in malnourished people 9374 × 10^3^/µL. The mean WBC at discharge from the Clinic was, in properly nourished patients 8.078 × 10^3^/µL, in patients at risk of malnutrition 7.504 × 10^3^/µL, and in malnourished patients 7.266 × 10^3^/µL. The mean CRP concentration upon admittance was 34.106 mg/dL in patients with normal nutritional status, 61.343 mg/dL in patients at risk of malnutrition, and 30.866 mg/dL in malnourished ones. The mean CRP concentration at the end of hospitalization was 24.519 mg/dL in patients with normal nutritional status, 18.800 mg/dL in patients at risk of malnutrition, and 7.757 mg/dL in malnourished people. It can be assumed that the obtained values result from factors other than nutritional level, which may be comorbidities or diseases that are the direct cause of hospitalization.

Due to the failure to meet the assumption of the equality of groups necessary for the parametric analysis of variance (ANOVA) (verification with the Chi-square test of equality, *p* < 0.001). Kruskal–Wallis tests were performed, which are its non-parametric equivalent. The results showed that there were statistically significant differences in adiponectin values in the individually analyzed groups (Table 2).

Post hoc tests showed differences in adiponectin concentration between the group of patients with normal nutritional levels and the group of malnourished patients (*p* = 0.004). No significant differences were found between the group of patients with normal nutritional status and those at risk of malnutrition (*p* = 0.392) and between patients at risk of malnutrition and patients with malnutrition (*p* = 1.000). The distributions of adiponectin measurement values in the groups are presented in Figure 1.

### 2.2. The Influence of Rapidly Changing Inflammation in the Course of Infection on Discussed Parameters

Table 3 shows the descriptive statistics and results of statistical tests of biomarkers depending on the use of antibiotics. The average uric acid concentration was comparable in the group of people using antibiotics during hospitalization due to bacterial infection (N = 36) and those not using them (N = 27) and amounted to 5.897 and 6.437 mg/dL, respectively. Concerning adiponectin concentration, no significant differences were observed between the described groups. The average concentration was 65.458 µg/dL in patients taking antibiotics and 58.443 µg/dL in patients not taking antibiotics during hospitalization (*p* = 0.361). Statistically significant differences were found for WBC and CRP measurements, both at the time of admission and upon discharge from the Clinic, but this is due to the characteristics of the groups discussed: patients taking antibiotics had high parameters of inflammation resulting from infectious diseases, which quickly decreased as a result of proper antibiotic treatment.

### 2.3. The Influence of Gender on Discussed Parameters

Table 4 presents the descriptive statistics and *p*-values of biomarker tests among patients divided by gender. According to the analyses, the uric acid concentration did not differ significantly in women (N = 26) and men (N = 37) and amounted to an average of 6.058 mg/dL and 6.178 mg/dL, respectively (*p* = 0.842). There was also no statistically significant difference in adiponectin concentration between women and men, which averaged 52.275 µg/dL and 69.494 µg/dL, respectively (*p* = 0.023). The WBC concentrations, both upon admission to the Clinic and at the end of hospitalization did not differ depending on gender. A higher average CRP concentration at admission and the end of hospitalization was found in women, which amounted to 78.723 mg/dL at the beginning of hospitalization and 32.368 mg/dL at the end of the stay, respectively. While in men the average CRP concentration on admission was 25.463 mg/dL and at discharge 10.402 mg/dL. These differences, however, may be consequences of the diseases that led to the hospitalization of individual study participants, which led to different responses to inflammatory parameters.

### 2.4. Biomarker Correlation Analysis

Spearman’s correlation coefficients were determined for the biomarkers analyzed, along with significance tests. The following graphs show the correlations among the entire study group, as well as by group of inpatients with normal nutritional status, at risk of malnutrition, and malnourished (Figure 2).

Among the entire study group, at the 0.05 significance level, significant correlations were found between WBC at the time of hospital admission and WBC upon discharge from the Clinic (R = 0.51), CRP at admission (R = 0.46), CRP at discharge (R = 0.35), and PCT on admission. CRP upon admittance is significantly correlated with CRP at discharge (R = 0.65) and PCT on admission (R = 0.47) and at discharge (R = 0.37). CRP upon discharge is also significantly correlated with PCT on admission and at discharge. Finally, PCT at the time of admission is correlated with PCT at discharge (R = 0.54) (Figure 2A). 

For patients with normal nutritional status, statistically significant correlations at the 0.05 level were shown between WBC measurements on admission and upon discharge from the Clinic (R = 0.68), as well as CRP measurements at the time of admission and discharge (R = 0.82) and PCT upon admittance (R = 0.72) (Figure 2B).

For patients at risk of malnutrition, significant correlations were found at the 0.05 significance level between WBC upon admittance and WBC at discharge (R = 0.45), CRP on admission (R = 0.5), CRP at discharge (R = 0.4), and PCT upon admittance (R = 0.42). In addition, WBC at discharge is statistically significantly correlated with CRP on admission and at discharge (R = 0.36 and R = 0.38, respectively). CRP at the time of hospital admission is significantly correlated with CRP at discharge (R = 0.63). CRP at discharge is statistically significantly correlated with PCT on admission (R = 0.4). Finally, PCT on admission is correlated with PCT at discharge (R = 0.55). (Figure 2C). Interestingly, there were no statistically significant correlations for biomarkers tested in malnourished patients (Figure 2D).

### 2.5. Effect of Antibiotic Therapy on the Change in WBC, CRP, and PCT Parameters between Admission and Discharge

To test the effect of antibiotic therapy on the change in WBC, CRP, and PCT parameters between admission and discharge, the differences in these measurements were determined and statistical tests were performed by groups defined by antibiotic intake. A negative value indicates a decrease in the value of the measurement at discharge relative to that taken at admission. Assuming the declared level of significance, there were no statistically significant differences in the values of the WBC, CRP, and PCT parameters (Table 5).

## 3. Discussion

A common reason for hospitalization, especially in older patients, is conditions involving a rapid increase in the inflammatory response. There are many causes of inflammatory reactions, including infectious diseases, rheumatological diseases, injuries, and cancer. During the treatment of patients with an inflammatory response, it is difficult to interpret commonly used inflammatory markers because they are characterized by the fact that their concentration increases in the general inflammatory response, which makes it difficult to specify what disease was the cause of the increase in inflammatory parameters [1]. For example, the treatment of infections in patients with advanced cancer can be carried out as effectively as possible. However, it is not always possible to achieve complete normalization of inflammatory parameters during hospitalization, because part of the inflammatory reaction will result from, for instance, the co-existing cancer [26]. The results of the present study confirm this phenomenon as substantial reductions in the WBC, CRP, and PCT (Table 5) values between the time of admission and discharge from the Clinic may be observed in most cases due to antibiotic treatment. However, the lack of statistically significant differences between the antibiotic and no antibiotic group suggests inflammation from causes different than bacterial infection. Therefore, it seems justified to search for further parameters measuring inflammation that could be used in clinical practice. The present study was conducted to demonstrate whether adipose tissue hormones, using the example of adiponectin, have the potential to act as a measure of rapidly changing inflammation when monitoring older patients in the course of infection. 

Older adults are a very fragile and specific group of patients. They are at high risk of bacterial infections, often present atypically, and have weaker immune responses, which results in high morbidity and mortality [27]. Moreover, because of lower prognostic values of routinely assessed parameters in comparison to younger adults, finding a reliable biomarker of bacterial infections would accelerate the proper diagnosis and effective treatment. The other important matter is an inappropriate prescription of antimicrobials to senile patients, which not only contributes to antibiotic resistance but also potential adverse effects, especially due to multiple chronic diseases and polypharmacy. 

The oxidation–inflammation theory of aging proposes that there is a close relationship between oxidative stress, inflammation, and aging. Aging is a loss of homeostasis caused by chronic oxidative stress, which mainly affects regulatory systems such as the immune system, nervous system, and endocrine system. This leads to activation of the immune system, inducing an inflammatory state, which creates a vicious cycle where chronic oxidative stress and inflammation reinforce each other. It leads to increased age-related morbidity and mortality [28]. There is a possible link between metabolic stress and oxidative stress in senile patients. It should be underlined that adiponectin plays a crucial role in maintaining metabolic balance and reducing oxidative stress related to obesity. Gradinaru et al. described significantly lower levels of adiponectin in senile patients with metabolic syndrome concomitantly with significantly higher levels of oxidative stress and cardiovascular disease risk markers [29]. Moreover, adiponectin levels were significantly positively associated with antioxidant capacity and significantly negatively associated with serum uric acid concentration. The results obtained in this study do not confirm the relationship between uric acid and adiponectin concentrations (Figure 2). The role of oxidative stress in acute and chronic infections, as well as in associated diseases, has been a topic of much debate. In bacterial infections, oxidative stress arises from altered metabolic pathways and has been linked to organ damage and the development of cancers. *Helicobacter pylori*, for instance, triggers enzymes that generate reactive oxygen species (ROS), such as spermine oxidase, and increases the expression of redox-regulated genes that are proinflammatory and potentially cancer causing, such as cyclooxygenase 2 [30].

Different effects of adiponectin in the human body have been already proven, including insulin-sensitizing, anti-atherosclerotic, and anti-inflammatory effects. This diversity is probably due to the existence of many isoforms of the hormone, which differ in their actions and the signaling pathways in which they participate. High concentrations of adiponectin are observed in many diseases caused by chronic inflammation. However, to understand and describe the effects of APN during the acute phase of inflammation, more research is necessary [9]. The most frequently described effect of APN is anti-inflammatory, even in diseases that do not specifically affect adipose tissue. It is so because receptors that may be influenced by adiponectin are located, for example, in the lungs. Therefore, therapies based on mechanisms involving APN may be used in the future in the treatment of inflammatory pulmonary diseases [25]. Naturally, due to the complex involvement of adiponectin in inflammatory processes, many researchers have been trying to identify the possibility of using it as a prognostic marker. So far, however, it has not been possible to link the dynamics of changes in adiponectin concentrations with ongoing acute inflammation. However, many studies have presented results suggesting that the measurement of a given protein can be used to detect and predict the course of some diseases, especially those in which there is chronic inflammation associated with obesity [31]. Determination of adiponectin concentration may probably be helpful in clinical practice in monitoring oncological diseases, diabetes, cardiovascular diseases, and Alzheimer’s disease [10]. A meta-analysis by Yang et al. showed that elevated adiponectin levels are an independent predictor of cardiovascular death and mortality in patients with coronary artery disease. It follows that the described hormone could probably contribute to identifying patients with coronary artery disease at high risk of death [32]. Additionally, the study by Güven and co-authors proved the usefulness of measuring APN in monitoring the course of breast cancer and diagnosing patients at risk of more aggressive disease. However, detailed research on large databases is necessary to introduce the described method into clinical practice [33]. Interestingly, the disturbance of the adiponectin-to-leptin ratio in the body may also play a role in the development of inflammation. This is especially true in patients with metabolic syndrome (MS). As shown by Frühbeck et al., the total concentrations of both laboratory-determined adiponectin and its multimeric forms were significantly lower in MS patients. Moreover, the quantitative adiponectin/leptin ratio in this group of people was significantly lower compared to the general population. It also should be noted that the study group showed increased levels of inflammatory markers, including CRP. This constitutes the basis for the conclusion that abnormalities in the functioning of adipose tissue, manifested as disturbances in the adiponectin/leptin ratio among MS patients, may increase the effects of oxidative stress and significantly increase the unfavorable effects of ongoing inflammation [34]. Another study by Chen et al. describes that concentrations of adiponectin, hs-CRP (highly sensitive CRP), IL-6, and the level of activity of antioxidant enzymes, i.e., superoxide dismutase (SOD), may also have a predictive value for the occurrence of MS in a patient. In this study recruited patients had significantly higher concentrations of the above-mentioned inflammatory markers, as well as lower adiponectin concentrations and reduced SOD activity. It can therefore be concluded that high concentrations of inflammatory markers combined with reduced adiponectin levels significantly correlate with a higher risk of MS [35]. In addition to the above-mentioned observations in senile MS patients, Ma et al., in their investigation, demonstrated a higher body mass index (BMI), higher blood glycemia, significantly increased CRP levels, and significantly reduced APN levels compared to the group of patients without MS [36]. The above-mentioned investigations demonstrate the validity of considering the concentration of circulating APN as a predictive marker for MS in older patients. In the present study, statistically significant differences were found for WBC and CRP measurements both at the time of admission and upon discharge from the Clinic. Notably, patients taking antibiotics had high parameters of inflammation resulting from infectious diseases, which quickly decreased as a result of proper antibiotic treatment (Table 3). However, no statistical differences were found for the level of APN between these two groups of patients. Some statistically significant correlations were found for inflammatory parameters such as WBC and CRP, but not for adiponectin (Figure 2). 

Some previous investigations indicate the role of APN in immunological response to infections. For instance, in the study by Behnes et al., it was shown that the level of the protein form of APN and the expression of the mRNA encoding it increase in response to the development of sepsis. Additionally, it has been proven that the deterioration of the patient’s general condition and severe sepsis are related to the increase in the concentrations of the mentioned substances in the blood. In laboratory tests, an increase in adiponectin and mRNA levels was observed as the disease progressed. Additionally, the study revealed that the group of patients treated with drotrecogin a (DAA)-activated protein C therapy was characterized by increased adiponectin expression during the ongoing systemic inflammatory reaction. Therefore, the study results confirm that adiponectin may play an important role in the pathogenesis of inflammatory processes in sepsis [37]. An attempt to demonstrate the relationship between the concentration of adiponectin and leptin and the presence of acute inflammation was made by Ekström and co-authors [38]. This condition was attempted to be induced in one of the study groups by administering a vaccination against Salmonella typhi. In the second group, measurements were performed after undergoing open-heart cardiac surgery. Therefore, two validated methods of inducing the inflammatory process were used. However, summarizing the results of adiponectin and leptin determinations in plasma, no changes were noticed in samples taken before and after the intervention. In addition, it was decided to determine the expression of adiponectin genes in omental and subcutaneous fat tissue, and no changes were detected there either. It follows that quantitative changes in the production and release of adiponectin probably do not depend on the processes of the acute inflammatory response. This phenomenon is consistent with the results obtained in this study. Interestingly, a slightly lower median level of APN is observed among inpatients suffering from infection in comparison to the “no antibiotic” group, but the lack of statistical significance does not allow conclusions at this stage of research. The results obtained so far may be considered preliminary and need further investigations, especially considering many possible comorbidities characteristic of the senile population. 

Adiponectin concentration is significantly influenced by body weight reduction, which deserves special attention in obese patients. It has been shown that reducing fat tissue while maintaining muscle mass in the weight-loss process in this group of patients leads to an increase in adiponectin concentration. This means that the assessment of serum adiponectin levels can serve as a predictor of the effects of weight loss in patients struggling with obesity [39]. The relationship between adiponectin concentration in the older population and their general condition, muscle mass, and the risk of mortality in this group of patients was investigated by Walowski et al. The results suggest that high levels of adiponectin are associated with lower muscle mass, reduced bone mineral content, and lower levels of fatty liver disease. This is most likely related to a decrease in the level of insulin-like growth factor 1 (IGF-1) in response to an increase in the concentration of adiponectin in the body [40]. The results of the above-mentioned studies are in agreement with the present investigation, which shows that a low nutritional status is associated with a higher concentration of adiponectin (Table 2). Significant differences in adiponectin concentration between the group of patients with normal nutritional status and the group of malnourished patients (*p* = 0.004) were observed (Figure 1). In contrast, Mądra-Gackowska et al. highlighted the influence of nutritional status according to the MNA on such adipokines as leptin and resistin [41]. No statistically significant differences between patients with normal nutritional status, at risk of malnutrition, and malnourished were found for APN. However, adiponectin concentration was inversely related to nutritional status, which was also observed in the current study. Indeed, the secretion of adiponectin by adipose tissue cells may be regulated by many factors, including diet and the patient’s overall nutritional status. Important factors modulating the synthesis of adiponectin are the elements of the diet and the percentage of its components. The most beneficial in this respect seems to be the Mediterranean diet and the DASH diet, the main element of which is the supply of optimal amounts of unsaturated fatty acids, but also fiber and polyphenols. Thanks to their positive impact on maintaining the optimal level of APN, the above-mentioned dietary formulas may therefore support the prevention of some lifestyle diseases containing an inflammatory component, i.e., cardiovascular diseases or cancer [42]. The inflammatory process is one of the fundamental mechanisms of obesity. In its course, inflammatory mediators are released from adipocytes into the bloodstream, which affect other body tissues. APN has anti-inflammatory properties and, in the case of obesity, it counteracts the negative effects of pro-inflammatory cytokines. An unfavorable effect of obesity is the gradual development of resistance to adiponectin, which over time leads to a significant reduction in the beneficial effects of this APN, which may result in an exacerbation of inflammatory processes [43].

This study revealed some statistically significant differences in inflammatory parameters and adipokine levels between groups established based on nutritional status or antibiotic use. However, there are some limitations of the present investigation. The measurement of serum APN was performed only once while repeating the measurement at the end of hospitalization could shed more light on the behavior of APN. The group of recruited patients, in whom the measurements were carried out, was relatively small. Presumably, for a larger groups, the results could be more accurate. For this reason, the results presented in this study cannot be fully applied to the entire population of older people. The focus was also only on demonstrating the relationship between adiponectin concentration and the course of acute inflammation. Completely different correlations can probably be found in patients struggling with chronic diseases involving the inflammatory process, for example cancer or rheumatological diseases. Therefore, it is necessary to conduct more studies involving older people to confirm the validity of using adiponectin as an acute inflammatory marker. 

## 4. Materials and Methods

### 4.1. Ethical Statement

The study was approved by the Bioethics Committee of the Collegium Medicum. L. Rydygier in Bydgoszcz. Nicolaus Copernicus University in Toruń on 14 December 2021. Consent number KB 675/2021.

### 4.2. Patient Recruitment

The study recruited 64 patients over 60 years of age, hospitalized in the Geriatrics Clinic of the University Hospital No. 1. Dr. A. Jurasz in Bydgoszcz in the period from July 2022 to April 2023, whose level of cognitive functioning allowed them to express informed consent to participate in the study. According to the organizational structure of the mentioned Clinic, patients can be hospitalized in two modes: planned, to perform a Comprehensive Geriatric Assessment (CGA), and urgent, when the health condition requires urgent hospitalization for general internal medicine reasons. Representatives of both groups took part in the study.

### 4.3. Nutritional Status Assessment

The Comprehensive Geriatric Assessment is a set of scales and tests used to assess independence in everyday functioning and to detect disorders that are part of the so-called Geriatric Giants such as falls, eating disorders, or frailty syndrome [44]. CGA is intended to indicate the patient’s problems and deficits and draw attention to their strengths, which may become a starting point for improving everyday functioning. As part of the assessment, the nutritional status is also checked. It is particularly important in estimating the body’s energy resources necessary to function while being exposed to multiple diseases [45]. A test that can be used to assess nutritional status is the Mini Nutritional Assessment (MNA). It includes the assessment of four basic parameters: anthropometric (weight, height, calf circumference, and arm circumference), food and fluid intake, general assessment (place of residence, physical activity, stressful situations or illness, medications taken, skin condition, neuropsychiatric disorders), and subjective assessment (self-assessment of health and nutritional status). During the assessment, points are awarded and then summed up: normal nutritional status is 24–30 points, a state at risk of malnutrition is 17–23.5 points, and malnutrition is a result below 17 points [46]. For the conducted research, all patients hospitalized both for a comprehensive geriatric assessment in a scheduled and urgent manner were assessed in terms of their nutritional status using the above-mentioned screening tool at the end of hospitalization, when their general condition allowed for assessment without a significant risk of the presence of factors interfering with the assessment, such as active, high inflammation. Depending on the number of points obtained according to the MNA, the study participants were divided into three groups:Group 0 (control) with normal nutritional level based on the MNA score.Group 1 at risk of malnutrition.Group 2 malnourished patients.

### 4.4. Taking into Account Comorbidities

The list of internal diseases that require hospitalization is long and varied. Patients who were able to give informed and voluntary consent to participate in the project were recruited for the study, with the following diseases: exacerbation of heart failure, complicated urinary tract infection, severe hyponatremia, severe anemia, pneumonia, acute kidney injury, decompensated diabetes, pulmonary embolism, and COVID-19. Since the number of diseases is large and they are often characterized by a different clinical course, further analyses took into account primarily the level of inflammation parameters and the use of any antibiotic therapy, bearing in mind that the analysis for specific disease entities individually may be irrelevant statistically with a limited number of research participants.

### 4.5. Evaluated Parameters

Uric acid is an example of a substance also involved in inflammatory reactions [47]. Its concentration was calculated at the beginning of patients’ hospitalization in the Clinic, as part of routine laboratory parameter determinations. Uric acid concentration was determined in patients’ serum using ELISA test. In analyses with other parameters, uric acid was included to demonstrate whether its concentration changes depending on changes in inflammatory parameters, such as WBC or CRP, or depending on the concentration of adiponectin, an example of an adipose tissue hormone.

Adiponectin was used as an adipokine that is relatively well known and can be commonly measured in medical care facilities. Serum APN concentration was also determined using ELISA test. Taking into account the fact that the studies involved patients with both increased inflammatory parameters and inflammation within the normal range, the level of adiponectin was determined once at the beginning of hospitalization, when discrepancies in inflammation parameters between individual patients were the largest.

In order to assess whether adiponectin concentrations change depending on the prevalence of acute inflammation, it was compared with other parameters of inflammation routinely assessed in hospitalized patients such as White Blood Cells (WBC), C-reactive protein, (CRP) and procalcitonin (PCT). These parameters are routinely measured in patients hospitalized at the Clinic upon admission and at the end of hospital treatment. The concentrations were determined in serum using ELISA test.

### 4.6. Consideration of Inflammation during Hospitalization

For further analyses, patients were also divided into two groups: patients with diagnosed infections who were treated with intravenous antibiotic therapy during hospitalization, and patients who did not require antibiotic therapy. The first group was always characterized by elevated inflammatory parameters that were subsequently decreased, while the second group could have chronically elevated inflammatory parameters (from comorbidities such as autoimmune diseases or cancer) or no inflammation.

### 4.7. Statistical Analysis

The Python programming language 3.8.10 (The Python Software Foundation, Fredericksburg, VA, USA) with the libraries pandas (1.4.3), matplotlib (3.1.3) and scipy (1.10.1) were used to perform the analyses. All analyses were performed assuming in advance a significance level of 0.05. After collecting the data described above, an attempt was made to demonstrate the correlation between the concentration of adiponectin, uric acid, WBC, and CRP on admission and at the end of hospitalization, and the nutritional status and the presence of acute inflammation, defined as inflammation caused by an infection requiring antibiotic therapy. Additionally, it was planned to demonstrate whether the mentioned parameters differed depending on gender. Verification was planned using the Chi-square test of equivalence and *p* < 0.001 to meet the assumption of equality of groups necessary for parametric analysis of variance (ANOVA). If verification is not possible, Kruskal–Wallis tests were planned to be performed. The Kruskal–Wallis test is used to compare two or more groups for a continuous or discrete variable. It is a non-parametric test assuming no particular distribution of the data and is analogous to the one-way analysis of variance (ANOVA) [48].

## 5. Conclusions

The present cross-sectional study revealed some statistically significant differences in inflammatory parameters and adipokine levels between groups of senile inpatients established based on nutritional status or antibiotic use. According to the results obtained, higher serum adiponectin levels are related to deterioration of nutritional status as significant differences in adiponectin concentration between the group of patients with normal nutritional levels and the group of malnourished patients were found. Despite bacterial infections being associated with oxidative stress rise, no statistically significant differences were found in the levels of APN in hospitalized patients diagnosed with acute infections who required antibiotic therapy compared to senile inpatients without diagnosed inflammation. Therefore, adiponectin may be a better biomarker of malnutrition in older adults than a parameter for monitoring the initial stage of acute inflammation. It should be noted that adiponectin was only measured once during hospital admission and commonly used markers of inflammation were tested throughout the hospitalization. Therefore, this study is preliminary and further research is needed. However, given the relationship between aging, infections, oxidative stress, and adipokines, it is possible that APN could be useful for monitoring later stages of the inflammatory response. As malnutrition is a serious and often underdiagnosed problem in the senile population, measuring adiponectin levels on hospital admission in the future may facilitate the identification of malnourished seniors. Nevertheless, further efforts should be made to search and validate potential biomarkers based on adipokines to translate them into clinical practice.

## Figures and Tables

**Figure 1 ijms-25-02016-f001:**
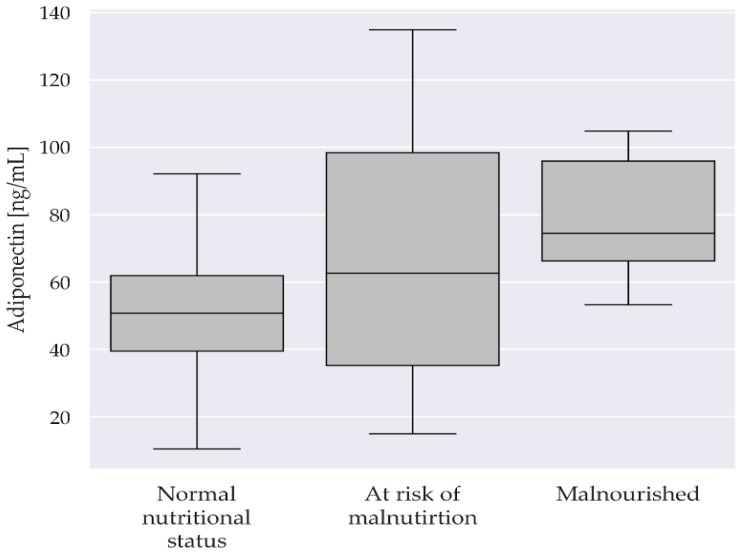
Boxplot of adiponectin divided by the MNA.

**Figure 2 ijms-25-02016-f002:**
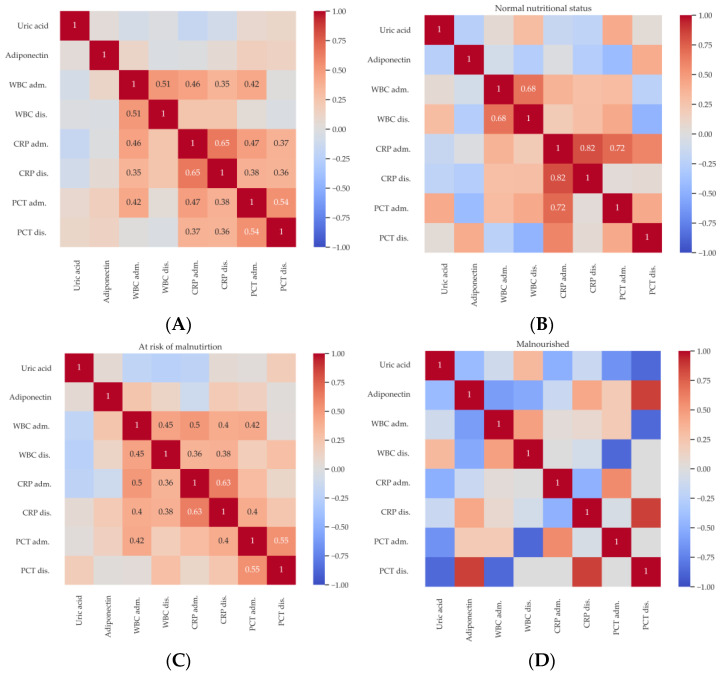
Spearman’s correlation diagram for all subjects (**A**), for patients with normal nutritional status (**B**), for the group at risk of malnutrition (**C**), and for malnourished patients (**D**). Correlations significant at the 0.05 level are marked in black or white.

**Table 1 ijms-25-02016-t001:** Descriptive statistics of biomarkers divided by the MNA.

Nutritional Status	Normal Nutritional Status							At Risk of Malnutrition							Malnourished						
Variable	N	Mean	Std Error	Q1	Median	IQR	Q3	N	Mean	Std Error	Q1	Median	IQR	Q3	N	Mean	Std Error	Q1	Median	IQR	Q3
Uric acid [md/dL]	24	5.47	0.35	4.18	5.40	1.93	6.10	32	6.21	0.45	4.35	5.60	3.95	8.30	7	8.04	0.91	7.05	8.10	1.75	8.80
Adiponectin [µg/mL]	24	50.23	3.89	39.52	50.85	22.48	62.00	33	67.79	6.14	35.25	62.67	63.17	98.42	7	79.63	7.38	66.38	74.45	29.60	95.98
WBC adm. [10^3^/µL]	24	9.20	1.16	5.84	8.08	4.84	10.68	33	10.30	0.95	6.70	8.44	5.92	12.65	7	9.37	1.42	6.49	9.84	4.84	11.32
WBC dis. [10^3^/µL]	23	8.08	0.82	5.95	7.29	2.36	8.31	31	7.50	0.46	6.20	7.40	1.90	8.10	7	7.27	0.84	5.76	7.14	3.57	9.33
CRP adm. [mg/L]	23	34.11	10.00	1.41	5.73	46.72	48.13	32	61.34	14.49	5.79	28.49	72.75	78.54	7	30.87	16.79	1.29	21.79	31.35	32.64
CRP dis. [mg/L]	20	24.52	6.93	1.89	6.79	40.90	42.78	31	18.89	3.43	4.31	11.24	23.98	28.29	6	7.76	4.03	2.06	3.06	6.11	8.17
PCT adm. [ng/mL]	12	0.08	0.02	0.05	0.06	0.07	0.11	26	3.05	1.59	0.09	0.18	0.55	0.64	5	0.19	0.12	0.02	0.08	0.13	0.15
PCT dis. [ng/mL]	9	0.07	0.02	0.03	0.05	0.06	0.09	20	1.20	0.96	0.06	0.10	0.20	0.26	3	0.02	0.003	0.02	0.02	0.01	0.03

**Table 2 ijms-25-02016-t002:** Kruskal–Wallis test results.

Variable	*p*-Value	Decision	Test Power
Uric acid [md/dL]	0.055	no differences in subgroups	0.981
Adiponectin [µg/mL]	0.033	statistically significant differences in subgroups	1
WBC adm. [10^3^/µL]	0.523	no differences in subgroups	0.9314
WBC dis. [10^3^/µL]	0.978	no differences in subgroups	0.5108
CRP adm. [mg/L]	0.196	no differences in subgroups	1
CRP dis. [mg/L]	0.458	no differences in subgroups	1
PCT adm. [ng/mL]	0.063	no differences in subgroups	0.9985

**Table 3 ijms-25-02016-t003:** Descriptive statistics and biomarker statistical test results by antibiotic use.

	No Antibiotic					Antibiotic					Statistical Tests		
	N	Mean	Standard Error	Median	IQR	N	Mean	Standard Error	Median	IQR	Type	*p*-Value	Test Power
Uric acid [md/dL]	36	5.897	0.345	5.450	2.750	27	6.437	0.513	5.900	4.300	U Mann–Whitney test	0.466	0.519
Adiponectin [µg/mL]	37	65.458	5.339	60.111	52.300	27	58.443	5.111	53.350	39.740	Student’s *t*-test	0.361	0.148
WBC adm. [10^3^/µL]	37	8.180	0.569	8.120	4.390	27	11.978	1.274	11.370	7.905	U Mann–Whitney test	0.010	1.000
WBC dis. [10^3^/µL]	34	7.052	0.439	6.535	1.873	27	8.500	0.682	7.960	2.320	U Mann–Whitney test	0.019	0.996
CRP adm. [mg/L]	35	25.466	8.543	3.480	28.07	27	76.746	14.907	49.140	118.510	U Mann–Whitney test	<0.001	1.000
CRP dis. [mg/L]	30	12.127	3.085	3.540	12.82	27	28.100	5.221	19.320	32.755	U Mann–Whitney test	0.004	1.000
PCT adm. [ng/mL]	19	0.213	0.065	0.080	0.145	24	3.215	1.724	0.140	0.552	Student’s *t*-test	0.130	0.326
PCT dis. [ng/mL]	13	0.089	0.029	0.050	0.060	19	1.243	1.008	0.100	0.200	Student’s *t*-test	0.354	0.149

**Table 4 ijms-25-02016-t004:** Descriptive statistics and *p*-values of biomarker tests among patients by gender.

	Male					Female					Statistical Tests		
	N	Mean	Standard Error	Median	IQR	N	Mean	Standard Error	Median	IQR	Type	*p*-Value	Test Power
Uric acid [md/dL]	37	6.178	0.386	5.600	3.400	26	6.058	0.465	5.600	3.300	Student’s *t*-test	0.842	0.054
Adiponectin [µg/mL]	38	69.494	4.883	62.434	42.880	26	52.275	5.402	44.868	34.973	Student’s *t*-test	0.023	0.629
WBC adm. [10^3^/µL]	38	9.135	0.816	8.260	4.272	26	10.728	1.123	10.480	7.600	U Mann–Whitney test	0.234	0.999
WBC dis. [10^3^/µL]	36	7.208	0.317	7.270	2.410	25	8.392	0.844	7.290	1.870	Student’s *t*-test	0.143	0.309
CRP adm. [mg/L]	36	25.463	6.044	9.170	35.788	26	78.723	17.279	48.125	126.482	U Mann–Whitney test	0.016	1.000
CRP dis. [mg/L]	33	10.402	2.080	6.150	9.160	24	32.468	5.975	21.175	48.272	Student’s *t*-test	<0.001	0.970
PCT adm. [ng/mL]	21	1.971	1.252	0.080	0.180	22	1.809	1.528	0.140	0.500	Student’s *t*-test	0.935	0.051
PCT dis. [ng/mL]	17	1.246	1.130	0.070	0.190	15	0.239	0.108	0.080	0.150	Student’s *t*-test	0.412	0.127

**Table 5 ijms-25-02016-t005:** Descriptive statistics and statistical test results of selected differences in biomarkers between admission and discharge from the Clinic by antibiotic use.

	No Antibiotic					Antibiotic					Statistical Tests		
	N	Mean	Standard Error	Median	IQR	N	Mean	Standard Error	Median	IQR	Type	*p*-Value	Test Power
WBC diff. [10^3^/µL]	34	−1.311	0.576	−0.195	2.003	27	−3.477	1.145	−2.27	7.120	U Mann–Whitney test	0.152	1
CRP diff. [mg/L]	30	−17.407	8.157	−0.605	14.855	27	−48.644	14.481	−22.86	85.84	U Mann–Whitney test	0.054	1
PCT diff. [ng/mL]	13	−0.192	0.083	−0.010	0.3	19	−2.778	1.992	−0.03	0.305	Student’s *t*-test	0.294	0.179

## Data Availability

Data are available on request due to privacy/ethical restrictions.

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
