# Peer review of "Preliminary Report on the Influence of Acute Inflammation on Adiponectin Levels in Older Inpatients with Different Nutritional Status"

_ijms, 2024, doi:10.3390/ijms25042016_

Round 1

Reviewer 1 Report

Comments and Suggestions for Authors

Dear Authors

 This paper provides some information on the relationship between adiponectin, a metabolic hormone from adipose tissue, and various health conditions, particularly in the context of inflammation and sepsis. Here are some key aspects and potential criticisms. The paper provides a comprehensive overview of the link between adiponectin and various conditions, including sepsis, aging, and metabolic syndrome. Research methodology and relevant factors such as diet and nutritional status that can affect adiponectin concentration are described in detail. The need for further research is recognized and limitations of the study, such as the relatively small sample of patients, are pointed out.

-Lack of clear connection to clinical practice: Although the work suggests a possible connection between adiponectin and certain health conditions, there is a lack of clear application of these findings in clinical practice. Better linkage with specific guidelines for treatment or preventive measures is needed.

-The paper emphasizes that the concentration of adiponectin depends more on the nutritional status than on the presence of inflammation. This may lead to questions about the reliability of adiponectin as a biomarker of inflammation, especially in situations where other comorbid conditions are present.

-The authors stated that there are limitations of the study, including a relatively small group of patients. These limiting facts reduce the generalizability of the results to the wider population.

-While the paper provides certain insights, there is a lack of a clear conclusion summarizing the main findings and their practical application.

-A clearer conclusion is recommended that would highlight key findings and implications for future research and practice.

Ultimately, the paper provides a foundation for further research, but could be improved by clarity in the presentation of results and their clinical implications.

Best regards

Author Response

Dear Authors,

This paper provides some information on the relationship between adiponectin, a metabolic hormone from adipose tissue, and various health conditions, particularly in the context of inflammation and sepsis. Here are some key aspects and potential criticisms. The paper provides a comprehensive overview of the link between adiponectin and various conditions, including sepsis, aging, and metabolic syndrome. Research methodology and relevant factors such as diet and nutritional status that can affect adiponectin concentration are described in detail. The need for further research is recognized and limitations of the study, such as the relatively small sample of patients, are pointed out.

We appreciate the Reviewer’s interest in our study, the significant effort to create this review, and all helpful suggestions. We addressed all the raised issues below and introduced substantial changes to our paper. You can find all changes throughout the paper marked in red.  

-Lack of clear connection to clinical practice: Although the work suggests a possible connection between adiponectin and certain health conditions, there is a lack of clear application of these findings in clinical practice. Better linkage with specific guidelines for treatment or preventive measures is needed.

We appreciate this valuable suggestion! In this respect, we supplemented the Introduction with information about the different roles of adiponectin in the human body and associated diseases (lines 89-100). Moreover, we also incorporated the description of adiponectin as a possible biomarker of different diseases (lines 131-143). When we planned our study we came across studies, in which the possible role of APN in viral or bacterial infections was shown and we cited these works according to a kind Reviewer’s suggestion. Finally, we specified why finding biomarkers of infections in senile patients is so important (nonspecific symptoms, weaker immune response, or lower prognostic value of routinely examined biomarkers)(lines 115-130). We are aware, that from the clinical point of view, it would be beneficial to specify a cut-off value for adiponectin, but taking into account the relatively small group of patients we did not decide to take this step for now. Our results may be considered preliminary and need further investigations on a larger population, which is why we also decided to change the title of our paper.

-The paper emphasizes that the concentration of adiponectin depends more on the nutritional status than on the presence of inflammation. This may lead to questions about the reliability of adiponectin as a biomarker of inflammation, especially in situations where other comorbid conditions are present.

Thank you for this comment! We agree! Indeed, at this stage of the study, we cannot consider adiponectin as a biomarker of inflammation in a course of bacterial infections. However, the studies involving older adults are challenging, because it is almost impossible to find an old individual without any ongoing disease. Quite the reverse, in most seniors comorbid conditions are present. So on the one hand, it is a pity that adiponectin seems not suitable to detect acute inflammation, but on the other hand, it may serve as a biomarker of malnutrition, which is one of Geriatric Giants and is often underdiagnosed. Moreover, it carries serious consequences, such as sarcopenia or depression, which lead to functional disability in the senile population. Despite the high incidence of malnutrition in hospitalized older adults and the availability of recognized methods to identify malnourished patients, screening is still carried out insufficiently and malnutrition is rarely identified in healthcare facilities. That is why high levels of APN if assessed routinely could attract the attention of medical professionals and lead to the faster diagnosis of malnutrition. Obviously, adiponectin as a biomarker of malnutrition in older patients needs further investigation.

-The authors stated that there are limitations of the study, including a relatively small group of patients. These limiting facts reduce the generalizability of the results to the wider population.

Thank you for this point of view. As we mentioned above, we changed the title of our paper, which now indicates that it is a pilot study. We also improved the description of the limitations of the study (lines 443-456).  

-While the paper provides certain insights, there is a lack of a clear conclusion summarizing the main findings and their practical application.

Thank you! We agree! The whole paragraph of “conclusions” was altered! (lines 548-567).

-A clearer conclusion is recommended that would highlight key findings and implications for future research and practice.

Thank you for raising this matter! As we mentioned above the conclusions section was rewritten and we hope that in the present shape, it better reflects the results of our study and implications for future research and practice (lines 548-567).

Ultimately, the paper provides a foundation for further research, but could be improved by clarity in the presentation of results and their clinical implications.

In this revision, we have made substantial changes according to the Reviewers’ comments. Moreover, your suggestions made us improve almost every line of our paper. So we hope that in the present shape or after further corrections it will be suitable for publication in the International Journal of Molecular Sciences. Anyway, we are open to further Reviewer’s suggestions!

Reviewer 2 Report

Comments and Suggestions for Authors

cDear Sirs, 

in my opinion, there is a methodology problem with this study. The authors have studied only CRP, WBCs and uric acid, whereas other inflammatory parameters, such as ferritin, IL-6, TNF-a, haptoglobins, procalcitonin and why not GGT, etc have not been taken into account. In addition, the discussion section, I cannot see any correlation between the results and the oxidative stress measurement, as stated in the title of this manuscript. Moreover, English language needs extensive editing. Therefore, I recommend rejection of this manuscript. 

Comments on the Quality of English Language

Dear Sirs, 

English language is too poor...

Author Response

Dear Sirs,

in my opinion, there is a methodology problem with this study. The authors have studied only CRP, WBCs and uric acid, whereas other inflammatory parameters, such as ferritin, IL-6, TNF-a, haptoglobins, procalcitonin and why not GGT, etc have not been taken into account. In addition, the discussion section, I cannot see any correlation between the results and the oxidative stress measurement, as stated in the title of this manuscript. Moreover, English language needs extensive editing. Therefore, I recommend rejection of this manuscript.

Thank you very much for the significant effort needed to create this review. We’ve found your suggestions very helpful in improving the quality of our manuscript. Below, please find the direct responses to your comments. All changes in the revised version of our manuscript are marked in red.

Indeed, we evaluated selected biochemical parameters important from the perspective of geriatric routine clinical practice. According to the Reviewer’s helpful suggestion, we also included procalcitonin, which was also examined for the majority of our inpatients, in statistical analysis. Parameters such as such as ferritin, IL-6, TNF-a, haptoglobins, and GGT are not determined routinely in the Geriatrics Clinic, where the recruited patients were hospitalized. The reasons for their not being used in clinical practice are primarily the costs associated with their implementation, as well as the implementation capabilities of individual laboratories, not all of which would be able to routinely and efficiently determine the mentioned parameters. In clinical practice in Poland, the concentration of white blood cells is primarily used to determine the degree of inflammation, as an element of a cheap and easy-to-perform blood count. Additionally, to detect developing, especially rapidly changing, inflammation, C-reactive protein (CRP) is used in clinical practice, which, despite its low specificity, is also used to assess the severity of some diseases associated with inflammation, such as rheumatoid arthritis. Therefore, their measurement provides not only information about the current intensity of inflammation but also assesses the chances of occurrence of certain diseases, therefore their use in routine treatment is justified. To assess the degree of inflammatory reaction in the course of a bacterial infection, procalcitonin (PCT) is used in clinical practice in Poland, the determination of which is also relatively cheap and quick and can be performed by virtually all laboratories. Bearing in mind that the study intended to attempt to use the obtained results in clinical practice, a decision was made to analyze parameters often used in routine treatment monitoring. Taking into account other inflammatory parameters mentioned by the Reviewer, despite the possibility of providing many interesting correlations, would have little relevance to clinical practice, where these parameters are checked relatively rarely.

We also agree that the title is exaggerated. Our study aimed to assess whether adiponectin may serve as a biomarker of acute inflammation in the course of bacterial infections.  We did not evaluate the parameters of oxidative stress, but we measured adiponectin levels. However, we are aware adiponectin levels and infections are strictly related to oxidative stress. That is why we decided to change the title to “Preliminary report on the influence of acute inflammation on adiponectin levels in older inpatients with different nutritional status” and we also implemented substantial changes to the discussion section. At this stage of research, adiponectin does not have the potential to act as a measure of rapidly changing inflammation in the course of bacterial infection. However, it may be a possible biomarker of malnutrition in older adults which is consistent with previous studies.

The Reviewers’ constructive criticism helped us make considerable changes to our paper. For instance, in the introduction, we added information linking infection with oxidative stress, multifaced roles of adiponectin in the human body, and metabolic diseases and introduced APN as a possible biomarker of various diseases, specified why finding biomarkers of infections in senile patients is so important (nonspecific symptoms, weaker immune response or lower prognostic value of routinely examined biomarkers).

In the results section, we presented descriptive statistics of PCT and corrected descriptions of correlation analysis and additionally, we examined differences in biomarkers (WBC, CRP, PCT) between admission and discharge from the Clinic by antibiotic use (Table 5).

The discussion section was also altered and in its present shape, we hope our results are better discussed in the context of other investigations and the state of the art. In particular, we emphasized the association of acute inflammation, bacterial infections, and adipokine levels with oxidative stress.

Moreover, our manuscript underwent extensive English editions according to the Reviewer’s suggestion.

We hope that in its present shape or after additional changes, you may consider our manuscript suitable for publication in the International Journal of Molecular Sciences. We are open to further Reviewer’s suggestions!

Round 2

Reviewer 1 Report

Comments and Suggestions for Authors

Dear authors,

Thank you for the accepted suggestions. My opinion is that this manuscript has no significant content and is not suitable for MDPI readers. The quality of the presentation is still at a low level.

Reviewer

Author Response

Dear Reviewer,

We would like to thank you for all the helpful suggestions we followed and for your effort to read carefully our paper once again. We are aware of the limitations of our study, which is why we retitled our paper to present our investigation to the readers as a preliminary study. Indeed, at this stage of the research, we cannot indicate adiponectin as a biomarker of acute inflammation. However, we found statistically significant differences in adiponectin levels depending on the nutritional status of older adults. Interestingly, a previous investigation, conducted in our Clinic by MÄ…dra-Gackowska and coworkers published in Antioxidants last year (doi: 10.3390/antiox12030569), revealed significant differences in leptin and resistin levels between the patients classified according to the MNA and the Geriatric Nutritional Risk Index (GNRI). Nevertheless, no significant differences were observed for adiponectin, which was achieved in our study with half the recruited population. Our results obtained encourage future studies with adiponectin in older adults. That is why we are planning a new investigation to be conducted in considering a larger number of participants. For the abovementioned reasons, we kindly ask you to consider publishing our work as a preliminary report.

Reviewer 2 Report

Comments and Suggestions for Authors

Dear Sirs,

the authors have made substantial improvement regarding their manuscript. However, there are some issues still to be resolved. In line 329, please delete that APN has also pro-inflammatory properties, as this statement would be misleading. In addition, the methods section should be the second section and not the fourth is this paper. The order is introduction, methods, results, discussion and conclusion. A conclusion section should also be written.

Author Response

We are grateful for the previous Reviewer’s criticism, which undoubtedly led to the improvement of our work. What is more, we appreciate your effort and careful analysis of our work this time!

In line 329 information about pro-inflammatory properties was deleted, according to a kind Reviewer’s suggestion.

In the case of section order, we agree and understand that it may be easier for the reader to get familiar with materials and methods first and after that with results/discussion and conclusions. However, we followed the Journal’s template and the order of chapters is specified there (1. Introduction; 2. Results, 3. Discussion; 4. Materials and Methods; 5. Conclusions). So please understand that we are going to maintain this order. The section of conclusions is after materials and methods, also according to the template.

Thank you very much!

Round 3

Reviewer 1 Report

Comments and Suggestions for Authors

Dear Authors,

I want to thank you for the exhaustive effort you have put in writing this paper. I read the paper again carefully and I am glad that the authors understand that there are limitations of the study. I agree that the current title is more appropriate for the study, and that at this stage of the research we cannot indicate that adiponectin is a biomarker of acute inflammation. Your results encourage future studies with adiponectin in the elderly. I support your plan to conduct new research.

After re-reading in detail, I think that this manuscript has the potential to be published in IJMS.

Best Regards